# Exploring the Patterns and Drivers of Urban Expansion in the Texas Triangle Megaregion

Jiani Guo and Ming Zhang *

School of Architecture, University of Texas at Austin, Austin, TX 78712, USA; jiani_guo@utexas.edu
* Correspondence: zhangm@austin.utexas.edu

**Abstract:** As the world becomes increasingly urbanized, it is vital for planners and policy-makers to understand the patterns of urban expansion and the underlying driving forces. This study examines the spatiotemporal patterns of urban expansion in the Texas Triangle megaregion and explores the drivers behind the expansion. The study used data from multiple sources, including land cover and imperviousness data from the National Land Cover Database (NLCD) 2001–2016, transportation data from the Texas Department of Transportation (TxDOT), and ancillary socio-demographic data from the U.S. Census Bureau. We conducted spatial cluster analysis and mixed-effect regression analysis. The results show that: (1) urban expansion in the Texas Triangle between 2001 and 2016 showed a decreasing trend, and 95% of the newly urbanized land was in metropolitan areas, especially at the periphery of the central cities; (2) urban expansion in non-metropolitan areas displayed a scattered pattern, comparing to the clustered form in metro areas; (3) the expansion process in the Texas Triangle exhibited a pattern of increased development compactness and intensity; and (4) population and economic growth played a definitive role in driving the urban expansion in the Texas Triangle while highway density also mattered. These results suggest a megaregion-wide emerging trend deviating from the sprawling development course known in Texas' urban growth history. The changing trend can be attributed to the pro-sustainability initiatives taken by several anchor cities and metropolitan planning agencies in the Texas Triangle.

**Keywords:** megaregion; urban expansion; spatiotemporal patterns; driving forces; the Texas Triangle

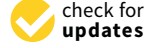



## 1. Introduction

Urbanization has long interested academia, policy-makers, and international agencies. One important aspect of urbanization pertains to urban land expansion. Urban land expansion (or urban expansion in short) is a process of creating urban land for the needs of urban population and activities [1]. According to the U.S. Census Bureau, about 80% of Americans lived in urban areas in 2018. Urban expansion is often accompanied by many ecological and environmental challenges [2], for example, ecosystem damage, traffic pollutions [3], climate change [4], and resource depletion. These challenges also adversely affect people and environment in both urban and rural areas [2]. Furthermore, massive and aggressive urban expansion has resulted in worsening social issues such as inequality, urban and rural poverty, and housing unaffordability [5].

There are many different perspectives to understand urban expansion. The neoclassical perspective of urban expansion emphasizes the role of free market in deciding the land to be developed for urban functions. This perspective holds that land price, transportation cost, income, and population distribution are predominant driving forces of urban expansion [6]. Researchers in this domain have developed sophisticated statistical models to explain and quantify the extent to which these forces drive urban expansion. On the other hand, the institutional perspective pays close attentions to the importance of institutional factors such as land use control, capital investments, and organizational capacities in the urban expansion process [7].

Common types of driving forces for urban expansion include changing geographic environment, economic development, population growth, technological advance, and public policies [8–12]. Geographic models and location choice theories have been developed and advanced widely in geography and urban economics. Geographers believe that humans tend to move to flat and warm places of rich and easily accessible resources. Location choice theories in urban economics state that industries would choose a location to minimize production costs and maximize profits [8]. In combination, urban expansion is most likely to happen in places satisfying both habitation preferences and economic wellbeing [13].

Economic development and its relationship to urban expansion has long been investigated. Jones and Kone found the positive relationship between per capita income and urbanization in the US, in late 20th century [14]. Lately, research also confirmed this relationship in other country settings. For instance, a study by Zeng et al. on the expansion in Wuhan, Hubei Province, China showed that gross domestic product (GDP) positively correlated with urban expansion at both micro and macro scale [9]. Scholars have also found that built environmental factors such as the distance to employment centers and/or major facilities (e.g., schools and hospitals) and the existing transportation network are also key contributors to urban expansion. Wang and Zhou used remote sensing data to fit logistic regression models to explore the urban expansion in Beijing-Tianjin-Hebei megaregion in China from 1984 to 2010 [15]. They found that "both local and tele factors statistically significantly affected the urban expansion process while the local factors played a relatively prominent role".

Public policies and governmental control play an essential role as well in affecting urban expansion outcome. One study in Puerto Rico has shown that "the ineffective plan of land development has left a high degree of urban sprawl in 40% of the island, where cities and towns appear typically surrounded by sprawl" [16,17]. Pham et al. discussed the different policy influences on urban expansion in four different cities worldwide [11]. They thought Shanghai's urban expansion patterns followed the policy guide of transition from mono-centric to multi-centric megaregion to decentralize the population and economic activities purposed by China's local and central government. With the continuous expansion of urban land, however, potential side effects emerge. To fight against the negative externalities, many local and federal governments have imposed restrictions on urban expansion. For example, the urban growth boundary initiated in 1979 in Portland, Oregon, was designed to limit urban development for resources. Research on this policy mainly focuses on measuring urban form [18] regarding urban sprawl and housing density. However, the urban expansion process under this particular urban growth boundary policy in the region is neglected. Several studies outside the U.S investigated the urban expansion patterns under local government policy. For instance, in Japan, the City Planning Act, which was promulgated in 1968, controlled the urban expansion, and their research confirmed the most urban expansion patterns only happened in limited places.

The United States has a long history of regional planning [19]. Extensive urban sprawl happening during the post-WWII development in the United States and many other countries have raised increasing concerns over the negative societal and environmental consequences [20]. Actions to counter sprawl have been taken, as some studies have found that urban expansion in major metropolitan regions has become more aggregated rather than ceaselessly expanding outwards [21]. Recent interests in megaregions call for improved understanding of urban expansion from a megaregional perspective, which motived this study.

With the rapid and foreseeable expansion trend, it is an urge for planners and policymakers to accommodate the shifting needs and to cultivate efficient land use via updated knowledge learned from analyses involving up-to-date data and comprehensive methods. As advocated in the planning field, managing urban expansion can be one key to balance sustainability's 3E triangle (i.e., equity, environmental protection, and economic development) and achieving sustainable development [22]. Thus, to generate a more

sustainable outcome, policy-makers should better understand the process and impacts of urban expansion and incorporate the findings in their policy guidelines. The Texas Triangle megaregion is one of the most populous and fast growing megaregions in the United States. Known for its affordable land price and business-friendly environment, the Texas Triangle is the future home to many major companies and populations [23]. Based on the context, we used remote sensing and U.S. census data to answer the following two questions:

1. What are the temporal and spatial patterns of urban expansion in the Texas Triangle, in terms of magnitude, clustering effects, and variations; and
2. What factors contribute to the urban expansion in this megaregion?

To answer those two questions, we first performed geospatial analysis to visualize the changing expansion patterns in the study area from 2001 and 2016 and analyzed the clustering effect during the period. Then, we fit a mixed-effect regression model to explore the relationship between the expansion intensity and socio-economic, transportation, institutional, and location factors. The findings of the study are expected to inform policy-making and strategic transportation investments for sustainable regional development.

## 2. Materials and Methods

### 2.1. Study Area

The Texas Triangle is one of the eleven megaregions in the continental U.S., identified by researchers from the University of Pennsylvania with RPA and the Lincoln Institute [24]. The megaregion lies within Texas and geographically encompasses four major metropolitan areas: Austin, Dallas-Fort Worth, Houston, and San Antonio. The Texas Triangle is connected by Interstate 45 (I-45), Interstate 10 (I-10), and Interstate 35 (I-35) (Figures 1 and 2). Most places in the Texas Triangle megaregion have a flat terrain while the west is hilly with elevation below 500 m. The climate in central Texas (including Austin, Waco and San Antonio) is semi-arid with average yearly precipitation from 530 mm to 890 mm [25]. The eastern region of Texas which is within the humid subtropical climate zone (including Dallas and Houston) has more than 1500 mm of annual precipitation.

We follow the definition of the Texas Triangle by Butler et al. [27] and Zhang et al. [28] with minor modifications. Using county as the geographic unit of analysis, megaregion is predominately defined by its economic and transportation connectivity, ecological and cultural similarity [28]. We replaced Delta County with Burnet County, in the original definition in Butler et al. study. The reason for adding Burnet County is concerning its inclusiveness in the Capital Area Metropolitan Planning Organization (MPO). Delta county is deleted because of its remote distance to the major highway and is not included in any MPO. Figure 2 presents a total of 66 counties and principal metropolitan areas, cities, and highways in the Texas Triangle megaregion.

The Texas Triangle had a population of over 21 million in 2018. Specifically, the Triangle megaregion has five of the top 20 most populous cities (Houston, Dallas, San Antonio, Austin, and Fort Worth) in the country. Moreover, there are four major Core-based Metropolitan Statistical Areas (CMSA) in this megaregion; they are Dallas-Fort Worth-Arlington, Houston-The Woodlands-Sugar Land, San Antonio-New Braunfels, and Austin-Round Rock.

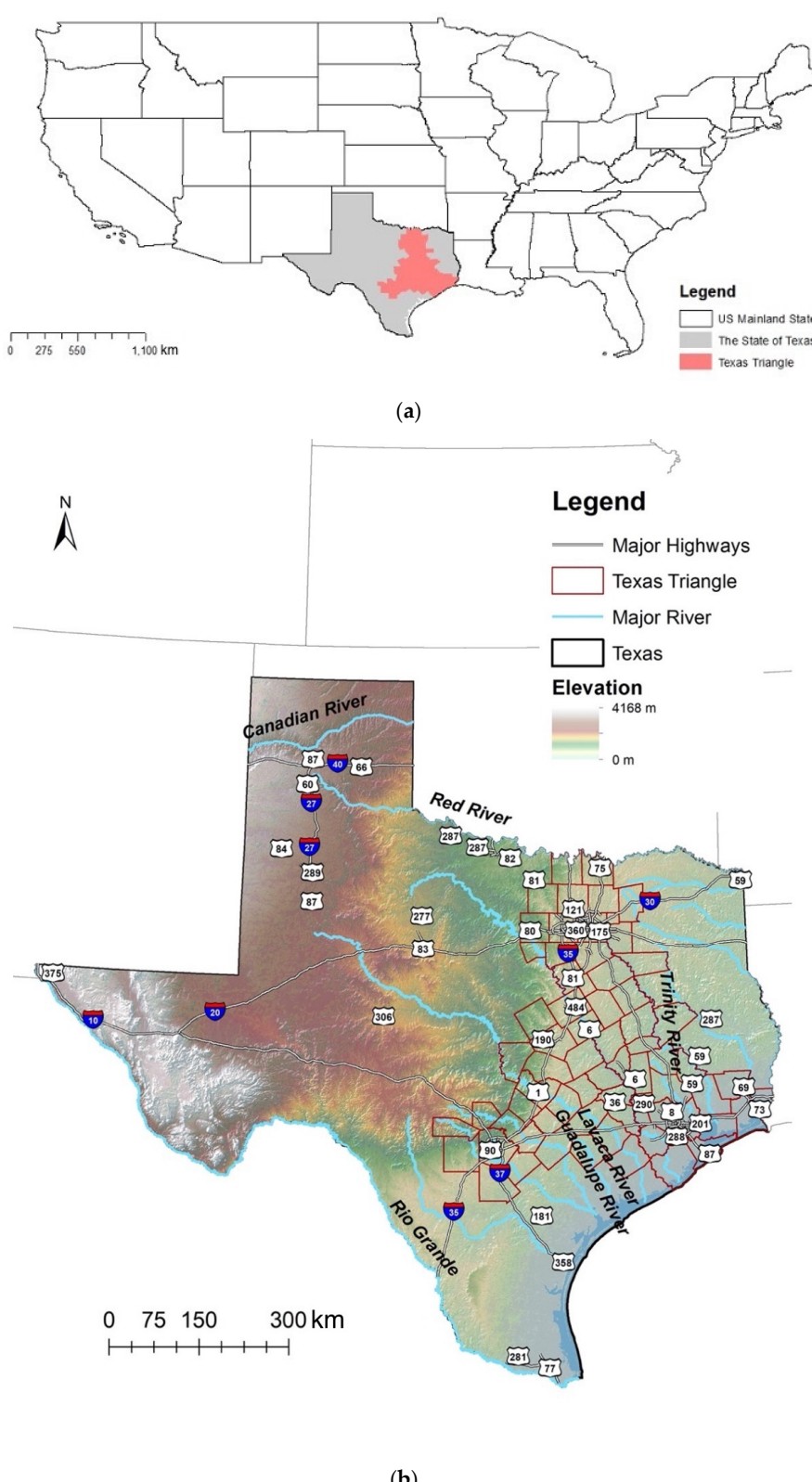

(**a**)

(**b**)

**Figure 1.** The Texas Triangle. (**a**): The location of the Texas Triangle in the U.S.; (**b**) the location of the Texas Triangle in the State of Texas. The elevation information is from the Texas Water Department Broad [26].

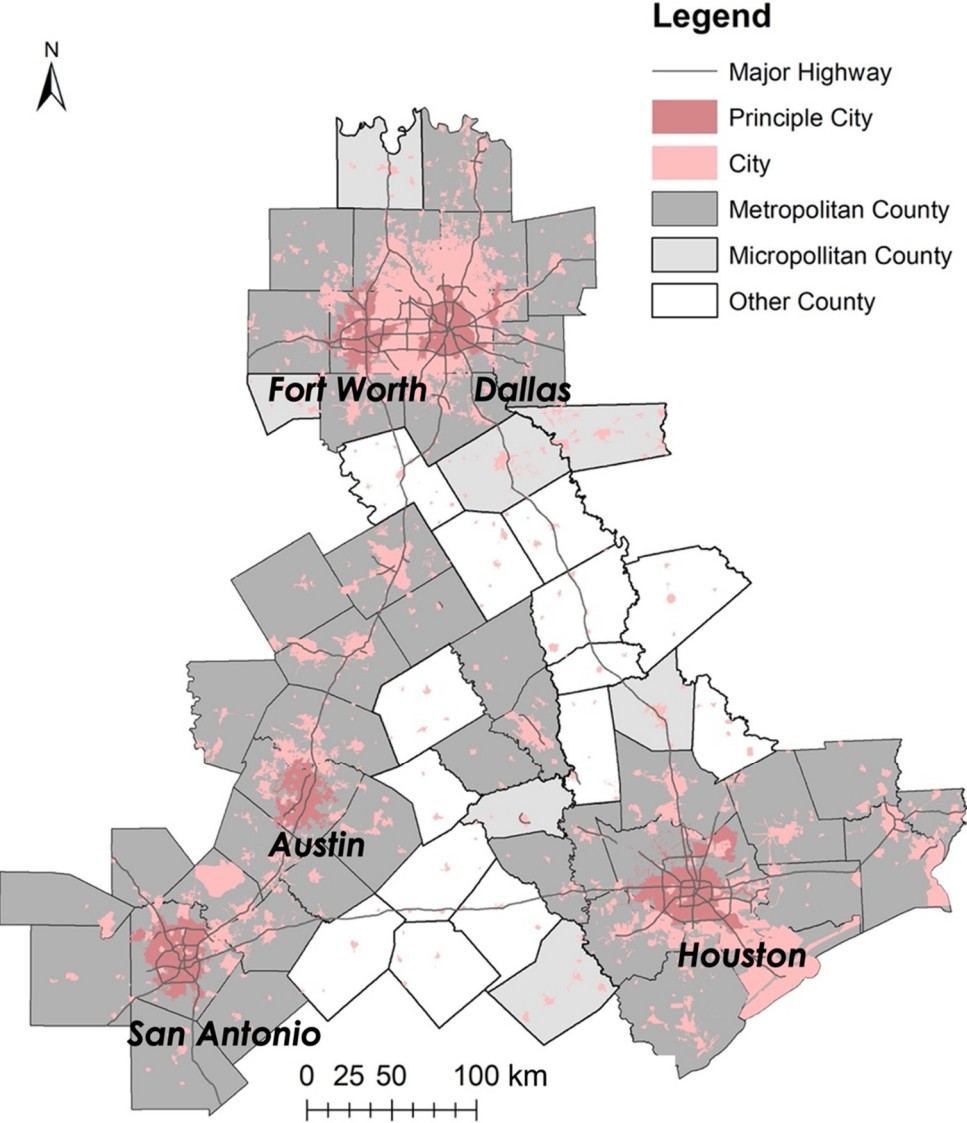

**Figure 2.** The Texas Triangle.

The Dallas-Fort Worth-Arlington MSA is the most populous metropolitan area in Texas and the fourth populous metropolitan area in the nation. It consists of 11 counties and a total area of 9286 square miles. This metropolis is home to 25 Fortune 500 companies, only behind New York City and Chicago. Houston-The Woodlands-Sugar Land is the second largest MSA in Texas and fifth most populous MSA in the U.S. This MSA includes nine counties (Harris, Fort Bend, Montgomery, Brazoria, Galveston, Liberty, Waller, Chambers, and Austin County) with a total area of over 10,000 square miles. Besides, this metropolitan area, is one of the fastest-growing MSA in the country. San Antonio-New Braunfels is an 8-county metropolitan area (Atascosa, Bandera, Bexar, Comal, Guadalupe, Kendall, Medina, and Wilson County), which covers a total area of 7387 square miles. As a famous historical city, the city of San Antonio is one of the top tourist cities in the U.S. Lastly, the Austin-Round Rock MSA includes six counties, Travis County, Bastrop County, Williamson County, Caldwell County, and Hays County. The Austin-Round Rock MSA is another rising metropolitan in which the population has increased from less than 300 thousand in 1970 to over 2 million in 2016 [29]. Austin was established in 1839 as the capital city of the Republic of Texas. The city is now a major education, technology, and economic center in the state, home to a flagship public university and world-renowned technology companies such as IBM (Endicott, NY, USA), Dell (Austin, TX, USA), and Apple (Los Altos, CA, USA).

### 2.2. Data

First, we retrieved the developed land from the National Land Cover Dataset (NLCD), U.S. Geographic Survey (USGS), to quantify the urbanized area. NLCD is a multi-year pre-prepared remote sensing data with a resolution of 30 m. The performance of the developed strategies and methods were tested in twenty World Reference System-2 path/row throughout the conterminous U.S. An overall agreement ranging from 71% to 97% between land cover classification and reference data was achieved for all tested areas and all years [30]. This remote sensing data has been used as a valuable data source in urban expansion research because of its broad and consistent area coverage and the virtue of being repeatedly updated regularly [31]. Rifat and Liu used NLCD and Coastal Change Analysis Program (C-CAP) datasets to study the urban expansion in the Miami Metropolitan area [32]. Terando et al. also used NLCD data to predict future urban sprawl in the Southern megapolis region [33]. In this research, the four categories of developed areas are treated as the urbanized area: developed open space, developed low density, developed medium density, and developed high density. The urbanized areas are calculated as the sum of the four types of developed land cover.

Moreover, we used the imperviousness data layer to retrieve the weighted urbanized area. Imperviousness data present impervious urban surfaces, representing the percentage of the developed surface. In the NLCD imperviousness dataset, each pixel is from 0% to 100%, where 80% to 100% pixels were classified as high intensity developed area. Several studies also used impervious information to measure the intensity of urban land [21,34]. We considered each pixel's imperviousness as an urbanized area's intensity weight. The weighted urbanized area is calculated as Equation (1):

$$\text{Weighted urbanized area} = \sum \text{pixel}_\text{i} * \text{impreviousness} \tag{1}$$

Because of the data availability, we used the NLCD data layers for the years of 2001, 2006, 2011, and 2016. Other data in this research correspond to the four years.

Highway data were collected from the Texas Department of Transportation (TXDOT), Roadway Inventory 2019. This dataset provides all the roadways records in Texas up to 2019, including length, width, road type, start date, and traffic volume. We selected major highways (including interstate highway, state highway and U.S. highway) and calculated their density at the county level as the transportation indicator. The highway density was measured as the total length of the highway dividing the total area of each county.

To calculate the indicator for innovations and technological advances, we used the patent data which were collected from the U.S. Patent and Trademark Office (USPTO). USPTO posts the number of patents that were registered in the corresponding year and registration county. We retrieved county-level patent data in corresponding years as an indicator of technology level in those years. Finally, other ancillary social-demographic data were from the Bureau of Economic Analysis (BEA), including population, employment, and GDP. Table 1 presents the descriptive data of urban land change and the key drivers by major MSAs in the Texas Triangle from 2001 to 2016.

### 2.3. Methods

To answer the first research question, we conducted an Anselin Local Moran's I cluster and outlier analysis. For the second research question, we performed a mixed-effect regression analysis to determine the factors related to the Texas Triangle's urban expansion. Detailed descriptions of the methods are as follows.

**Table 1.** Descriptive data in the Texas Triangle, 2001–2016.

| Metropolitan | Area (km²) | | | Population | | Employment | | GDP | | Patent | | Highway Length | |
|---|---|---|---|---|---|---|---|---|---|---|---|---|---|
| | Total | Urbanized Area in 2016 | % Change Since 2001 | 2016 (Millions) | % Change Since 2001 | 2016 (Millions) | % Change Since 2001 | 2016 ($ Billion) | % Change Since 2001 | 2016 | % Change Since 2001 | 2016 (km) | % Change Since 2001 |
| Austin-Round Rock-Georgetown | 11,085.37 | 1612.37 | 25.86 | 2.06 | 56.06 | 1.38 | 60.47 | 124.22 | 102.94 | 2701.00 | 55.86 | 1469.35 | 137.23 |
| Beaumont-Port Arthur | 6189.38 | 750.96 | 6.71 | 0.39 | 2.63 | 0.21 | 8.39 | 24.83 | 6.43 | 34.00 | 17.24 | 830.40 | 17.28 |
| College Station-Bryan | 5525.32 | 408.17 | 18.93 | 0.25 | 32.76 | 0.15 | 39.81 | 12.91 | 79.67 | 67.00 | 45.65 | 508.76 | 42.44 |
| Dallas-Fort Worth-Arlington | 23,328.57 | 5528.23 | 18.69 | 7.19 | 34.90 | 4.79 | 38.32 | 432.21 | 55.19 | 3028.00 | 42.09 | 6038.74 | 24.78 |
| Houston-The Woodlands-Sugar Land | 24,459.42 | 5531.00 | 21.98 | 6.81 | 41.29 | 4.04 | 40.40 | 446.78 | 50.18 | 3184.00 | 78.98 | 2959.86 | 56.31 |
| Killeen-Temple | 5554.46 | 498.58 | 17.00 | 4.16 | 30.82 | 0.22 | 26.61 | 15.81 | 47.07 | 21.00 | 5.00 | 452.14 | 67.33 |
| San Antonio-New Braunfels | 19,090.44 | 2162.71 | 17.66 | 2.42 | 38.44 | 1.41 | 41.86 | 108.63 | 62.50 | 413.00 | 73.53 | 3036.14 | 19.73 |
| Sherman-Denison | 2536.11 | 211.95 | 5.14 | 0.13 | 15.73 | 0.07 | 16.74 | 4.45 | 44.87 | 16.00 | −33.33 | 485.67 | 10.31 |
| Waco | 4750.26 | 412.30 | 7.48 | 0.26 | 11.41 | 0.16 | 22.15 | 11.12 | 40.66 | 17.00 | 54.55 | 436.64 | 29.47 |
| **The Texas Triangle** | **117,767.30** | **18,030.67** | **18.28** | **20.34** | **37.34** | **12.64** | **40.13** | **1195.98** | **56.13** | **9539.00** | **58.11** | **17,510.89** | **34.52** |

### 2.3.1. Anselin Local Moran's I Cluster and Outlier Analysis

We identified hot-spot clusters and spatial outliners of urbanized land at the Census tract level through the Anselin Local Moran's I statistic [35]. This method is widely used in many fields, such as economics [36], demographics [37], and geography [38]. The Anselin local Moran's I cluster and outlier analysis (cluster analysis) is adopted because of its ability to capture the spatial patterns in not only their general but also abnormal trends. We used this method in our study to categorize four types of spatial clusters of urban expansion in the Texas Triangle. If the Local Moran's I test statistic turns out to be positive, this area belongs to a statistically significant cluster of either a high value (a high-high cluster) or a low value (a low-low cluster). On the contrary, if the test statistic is negative, this area is an outlier of either high value surrounded by low-value areas (a high-low outlier) or otherwise (a low-high outlier). We tested the absolute increase area of urbanized land at the census tract level in 2001 to 2006, 2006 to 2011, and 2011 to 2016 and to see if a place is the hotspot of urban expansion, or if this place is the outlier with the abnormal increasing urbanized land while its surrounding areas are not.

### 2.3.2. Regression Analysis of the Driving Forces of Urban Expansion

We estimated mixed-effect regression models to measure the relationship between urban expansion and its potential driving forces. The mixed-effect regression model can cancel out the unobserved the error from different geographic entities and other potential error. The time variables were fixed to test its relatively growth in different periods.

The dependent variable urban expansion is measured by the percentage of urbanized area and the percentage of the weighted urbanized area in the county. Besides selecting the urbanized area, we added the weighted urbanized area to model the intensity growth in this megaregion. The comparison of absolute urbanized area and weighted urbanized area can depict a more comprehensive urban expansion process beyond the horizontal land cover changes.

To model the urban expansion, we selected six widely discussed variables in the literature that can capture the most context at a higher level, such as the megaregion level used in this study. The details and rationales are elaborated and explained in the following paragraphs. Independent variables were categorized into five types, as is shown in Table 2.

**Table 2.** Selected major drivers of urban expansion.

| Variable Category | Variable | Description | Sources |
|---|---|---|---|
| Social demographic factors | Population | The total population in the county | Bureau of Economic Analysis |
| Economic factors | Jobs | Employment in the county | Bureau of Economic Analysis |
| | GDP | GDP (millions) in the county | Bureau of Economic Analysis |
| Intellectual and technology innovation | Patents | Number of patents in the county | U.S. Patent and Trademark Office |
| Transportation infrastructure | Highway Density (kilometer per square kilometers) | The ratio of the length of highway to the total area in the county | TxDOT Roadway Inventory 2019 |
| Institutional factor | 1 if the county is part of a Metropolitan area; 0 otherwise | If this county is within/out of a Metropolitan area | The U.S. Census Bureau |

We selected the population as one of the major predictors. It is indubitable that population growth and land expansion are two inseparable aspects of the urbanization process [39]. Research has shown that population migration from rural to urban areas is a major driving force of urban expansion [40]. Therefore, it was our expectation that the most important driving force of urban expansion was population growth, measured as the total population in the county in the corresponding years.

Second, we included the economic indicators in the model because urban economists think urban expansion results from the market and economy agglomeration and expansion [6]. Economic development and increasing economic activities have accelerated the urban expansion process in recent decades. On the other hand, urbanization also may, in return, promote economic development. The economic advantages in the urbanized

area further attract more population and migration from rural to urban areas. Hence GDP and the number of employments widely serve as two indicators to measure the economic development in different counties.

The number of patents in the county was selected as an indicator of technology innovation. Technology development is wildly considered a significant factor of the prosperity of a region. The first technological revolution in the later 18th century in the United Kingdom is also the time the urbanization began. The second and the third technological revolution in the U.S. accelerated the urbanization process and urban expansion. From industrialization and informatization, technology innovation is always one of the central forces pushing the urban expansion process [41]. Friedman once argued that the technology is one of the reasons that the geographical location is less important nowadays [42]. In this research, the number of patents in the county is used as a proxy of technology innovation, as was used in Florida's research [41].

Highway density was chosen to represent the capital investments in transportation infrastructure. Early from the bid-rent theory, the distance to major transportation facilities is essential to location choice [43]. Later on, Dr. Adam's four stages model further emphasized how transportation infrastructure can shape the urban form and lead to urban expansion in different phases [44]. Moreover, transportation density is also an important indicator of built environments, influencing the urban expansion process. Therefore, the highway density is calculated to measure the supply of transportation infrastructure in each county.

The institutional perspective focuses on institutional or municipalities' role in the urban expansion [7]. The governmental policy is another factor for urban expansion. Due to the intricate and fragmented municipalities and governmental systems at the megaregion level, we considered being in a metropolitan area an institutional factor to investigate whether a county belongs to a larger administrative unit will make a difference in their urban land expansion. A metropolitan statistical area, defined by the U.S. Office of Management and Budget, is region with a principal city and its periphery containing more than 50,000 population. A micropolitan statistical area similarly is a place with population between 10,000 and 50,000.

The initial status (in the year 2001) of the urbanized area in each county varies; however, the growth rates of urbanized areas are relatively similar among different counties. Therefore, we did not use a growth model because of the low variation in slopes. In this case, we fixed the time effects, setting the initial year, 2001 as the baseline, while setting the geographic entities, that is each county, as random effects. The choice and form of variables are presented in Table 3.

The form of the equation is shown in Equations (2) and (3):

$$
\begin{aligned}
\text{lglandpct}_i = \beta_{0i} \quad &+ \beta_{1i}*\text{metro} + \beta_{2i}*\text{lgpop}_i + \beta_{3i}*\text{lgjob}_i + \beta_{4i}*\text{cpatent}_i \\
&+ \beta_{5i}*\text{lggdp}_i + \beta_{6i}*\text{lghwden}_i + \beta_{7i}*\text{year}_{2006i} + \beta_{8i} \\
&*\text{year}_{2011_i} + \beta_{9i}*\text{year}_{i_{2016}} + U_i + \varepsilon_i
\end{aligned}
\tag{2}
$$

$$
\begin{aligned}
\text{lgimppct}_i = \beta_{0i} &+ \beta_{1i}*\text{metro} + \beta_{2i}*\text{lgpop}_i + \beta_{3i}*\text{lgjob}_i + \beta_{4i}*\text{cpatent}_i + \beta_{5i} \\
&*\text{lggdp}_i + \beta_{6i}*\text{lghwden}_i + \beta_{7i}*\text{year}_{2006i} + \beta_{8i} \\
&*\text{year}_{2011_i} + \beta_{9i}*\text{year}_{2016_i} + U_i + \varepsilon_i
\end{aligned}
\tag{3}
$$

**Table 3.** Description of the variables in the mixed-effect regression.

| Dependent Variable | |
| --- | --- |
| lglandpct | Logarithm form of percentage urbanized land (%) in the county |
| lgimppct | Logarithm form of percentage weighted urbanized land (%) in the county |
| **Independent Variable** | |
| metro | =1 if the county is in a metropolitan area |
| lgpop | Logarithm form of the population in the county |
| lgjob | Logarithm form of jobs in the county |
| cpatent | =0, if the number of patents is 0 in the county; =1, if the number of patents is between 1 and 5 in the county; =2, if the number of patents is between 6 and 100 in the county; =3, if the number of patents is between 101 and 1000 in the county; =4, if the number of patents is above 1001 in the county |
| lggdp | Logarithm form GDP (in millions of dollars) in the county |
| year | 2001, 2006, 2011, 2016 |
| lnhwden | Logarithm of the length of the highway in the county (km)/Area of the county (km$^2$) |

## 3. Results

### 3.1. Spatiotemporal Patterns of Urban Expansion in the Texas Triangle

Figure 3 shows the change in urbanized area from 2001 to 2016 in the Texas Triangle. The newly developed urban land is mainly concentrated in the periphery of the major metropolitan area evident in the figure. The newly developed urban area in other counties presents scattered patterns. Additional cluster analysis further confirms that there are no cluster effects in those counties.

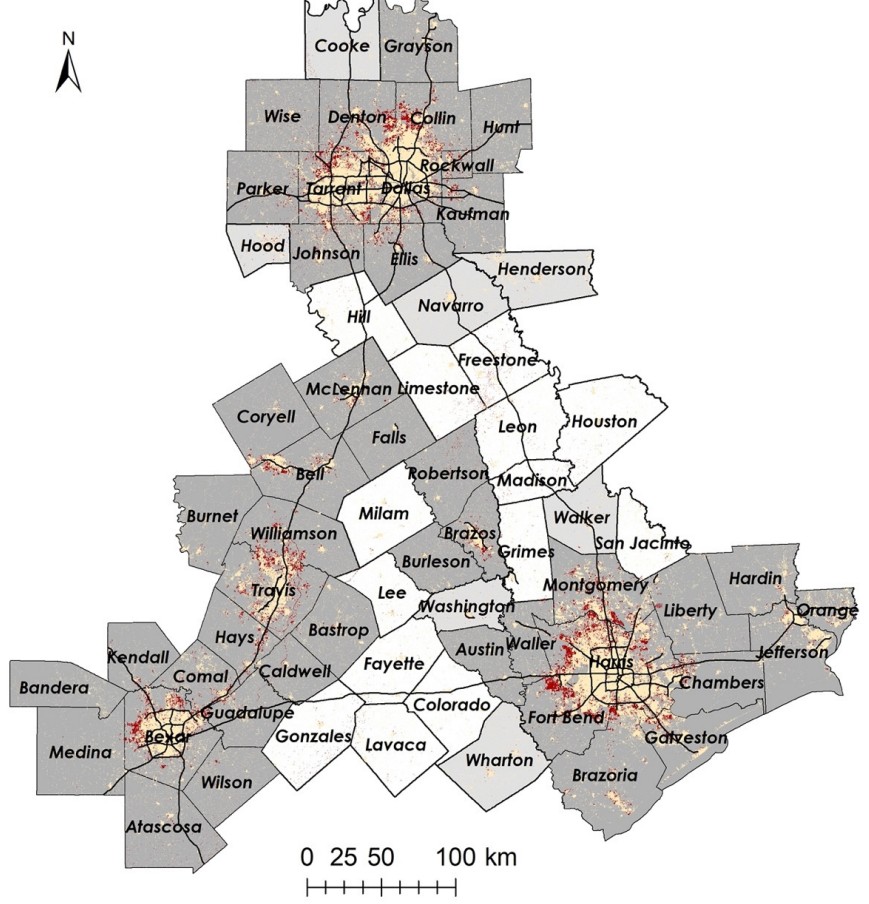

**Figure 3.** Urbanized Area in the Texas Triangle.

Figures 4 and 5 show the spatial and temporal patterns of urban expansion in the Texas Triangle during the three time periods. The results, first, illustrate higher growth rates in metropolitan counties than other counties. From 2001 to 2016, the urbanized area has increased by 2887 km$^2$, while 95% of those expansions occurred in metropolitan areas. Moreover, Figure 5 shows the decreasing growth rate over time in the Texas Triangle.

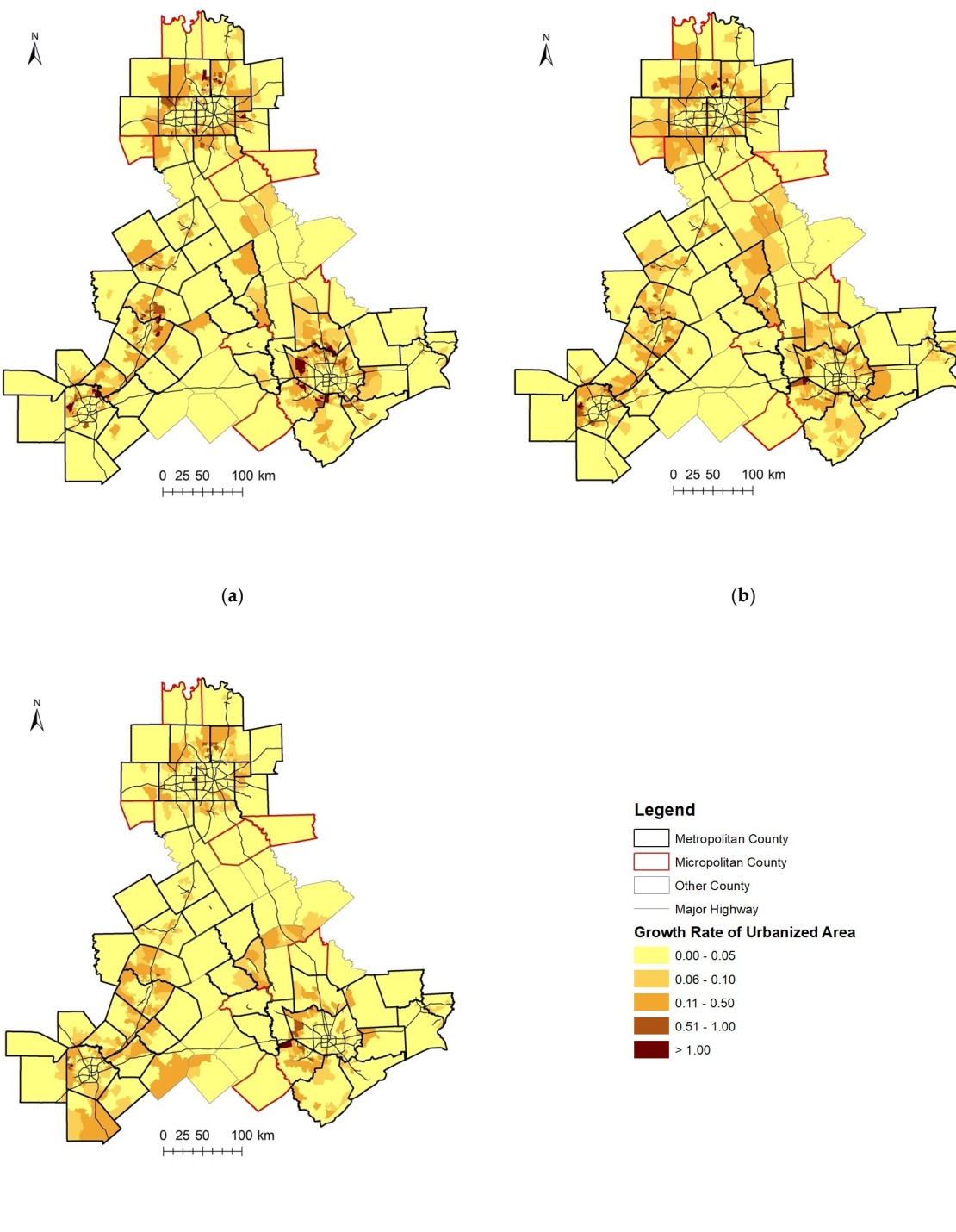

(**a**)                                                                                                (**b**)

(**c**)

**Figure 4.** Maps of Urbanized Land Growth Rate in the Texas Triangle from 2001 to 2016: (**a**) 2001–2006; (**b**) 2006–2011; (**c**) 2011–2016.

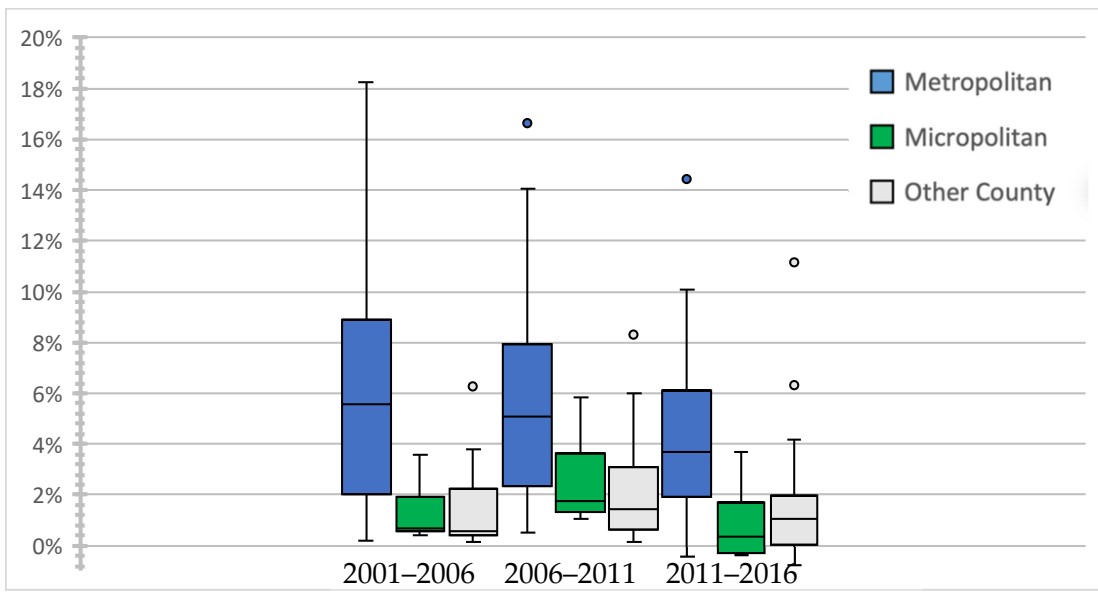

**Figure 5.** Urbanized land growth rate in different geographic area in the Texas Triangle.

Maps of growth spatial clusters and outliers in the megaregion (Figure 6) illustrate the spatial characteristics of urban expansion from 2001 to 2016 in the Texas Triangle. To start with, counties in four major metropolitan areas presented more high-high clusters of urban growth than micropolitan counties and other counties. The four major MSAs commonly exhibited patterns from principal cities outward: low-low cluster, low-high outlier, high-high cluster, and high-low outliers. This pattern was location-irrelevant in all periods. Specifically, central counties in major metropolitan areas, i.e., Travis county in Austin MSA, Dallas and Tarrant County in Dallas MSA, Bexar County in San Antonio MSA, and Harris County in Houston MSA, had a relatively low increase rate. Principal cities, Austin, San Antonio, Houston, Dallas, and Fort Worth, all presented significant low-low growth clusters, while the high-high cluster aggregated in slightly different localities at different time intervals. However, the urban expansion has an extraordinary intensity in the periphery area around the central county of metropolitan areas like Fort Bend County in Houston metropolitan, Rockwall County in Dallas metropolitan, and Williamson county in Austin metropolitan. Moreover, the urbanized land growth rates decreased over time, and the growth moves farther away from the principal cities. Interestingly, many census tracts with comparatively high growth rates are on the north side of metropolitan areas.

Besides the commonalities, four major metropolitan areas exhibited different urban expansion patterns. The Houston MSA shows relatively consistent and intensive expansion patterns in the outer ring on the north and east sides. This pattern might relate to its adjacency to the ocean to the south. Whereas the San Antonio MSA presents a relatively low expansion pace overall, and those expansions are concentrated on the north side. The Austin MSA has the fastest growth rate in the Texas Triangle. While development on the north Austin MSA dated back from 2001, the south side started to consume significantly more land as urbanized land from 2006. Lastly, the Dallas MSA has a high growth rate in the periphery places around Dallas and Fort Worth from 2001 to 2006. Then, from 2006 to 2011, the urbanized land grew primarily on the north and southwest in the metropolitan region. In the last time period, the urban land mainly expanded only on the north side.

In other smaller metropolitan areas, urban expansion patterns are slightly different from the major ones. Growth rates in those areas are generally slower, except for the College Station metropolitan. Besides, there are no prominent spatial clusters of high or low values in urbanized land growth. It is worth noting that, in the connecting MSA counties between four major MSAs, the urban expansion patterns are different over time. Specifically, counties between Dallas and Houston experienced a sizeable urban expansion

in the first two periods, whereas more expansion was found between Austin and San Antonio in the third.

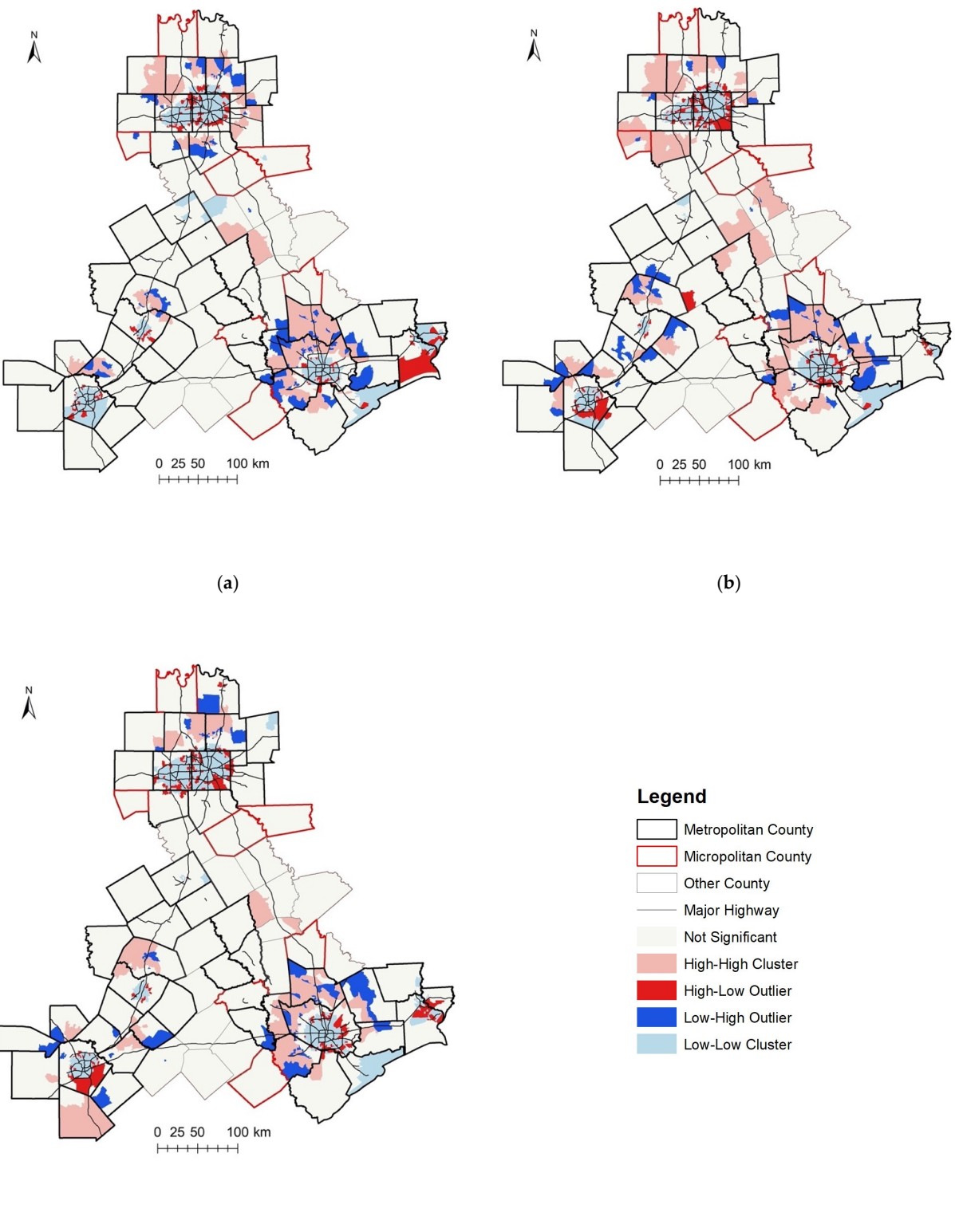

(**a**)

(**b**)

(**c**)

**Figure 6.** Spatial cluster and outliners of newly developed urbanized land (**a**) 2001–2006; (**b**) 2006–2011; (**c**) 2011–2016.

However, in non-metropolitan counties, the scale of urban expansion is relatively small. Moreover, from the cluster analysis, there are no significant urban expansion clusters in all periods, indicating a scattered expansion pattern in those counties.

To sum up, the newly developed urbanized land concentrates mainly on the periphery of core cities of major metropolitan areas. Overall, the urbanized land in the metropolitan area has expanded to suburbs, but the growth rate has declined over time. Even though more undeveloped land has changed to urbanized land during the whole period, the results above show that the expansion happened in a more aggregated manner instead of randomly sprawling in the metropolitan counties. Those patterns accord with the urban expansion patterns in major urban areas in the US Great Plains from 2000 to 2009 [45] with showing a compact development trend. Comparably, the growth in non-metropolitan counties is slower and more scattered.

### 3.2. Regression Results

Table 4 presents the result of the regression analysis. There are in total 262 observations, representing 66 counties in 4 years (the Milam County had no highway in 2001 and 2006 and therefore omitted in the regression models). The overall $r^2$ in the two models are 0.86 and 0.66, which explains most variations by the models.

**Table 4.** Results of the mixed-effect regression.

| Dependent Variable | Lglandpct | Lgimppct |
| --- | --- | --- |
| metro | 0.0292 | 0.0325 |
| | (0.39) | (0.49) |
| lgpop | 0.3794 *** | 0.143 ** |
| | (8.25) | (2.38) |
| lgjob | 0.0150 | −0.0375 |
| | (0.35) | (−0.66) |
| lggdp | 0.0342 *** | 0.0722 *** |
| | (3.43) | (5.16) |
| cpatent | 0.0065 | 0.0117 * |
| | (1.54) | (1.95) |
| lghwden | 0.0105 ** | 0.0215 |
| | (2.68) | (1.51) |
| Year | | |
| 2001 | 0 | 0 |
| | (.) | (.) |
| 2006 | −0.00047 | 0.0201 *** |
| | (0.2) | (3.01) |
| 2011 | 0.0074 | 0.0442 *** |
| | (1.61) | (3.01) |
| 2016 | 0.0041 | 0.0556 *** |
| | (0.72) | (6.36) |
| _cons | −2.724 *** | 1.142 *** |
| | (−14.13) | (5.33) |
| $\sigma_u$ | 0.26542 | 0.20912 |
| $\sigma_e$ | 0.02482 | 0.03268 |
| ρ | 0.99170 | 0.97614 |
| overall $r^2$ | 0.8644 | 0.6553 |
| N | 262 | 262 |

Note: significance level: ***: $p < 0.001$; **: $p < 0.01$; *: $p < 0.05$.

First of all, the models coincide with previous literature and show that population growth statistically significantly leads to greater expansion. Surprisingly, albeit 95% of the expansion in the Texas Triangle happened in the metropolitan areas during the entire period, being a metropolitan county shows no statistically significant advantages than other counties in both models. The result implies that metropolitan setting does not explain the trend of urban expansion in the Texas Triangle megaregion.

In terms of the time variable, compared to 2001, while controlling for other factors, there is no significant growth of the urbanized area from 2001 to 2016. However, the weighted urbanized area shows totally different results, and all-time variables are highly positively correlated to weighted urban areas. Unlike economic development or GDP, where we assume there might be natural growth because of productivity or efficiency improvement, the total urbanized area shows no such natural growth in the Texas Triangle from 2001 to 2016 while controlling population, economic development, and other factors. Nevertheless, it is worth noting that the intensity of urban areas has such growth from 2001 to 2016 when controlling other variables. That is to say, from 2001 to 2016, the urban expansion process in the Texas Triangle is more compact rather than low-density development. From this aspect, it is possible to control the urban growth if policies are controlling for population and transportation infrastructure. Moreover, it shows the results of promoting compact development.

As for economic factors, the models reveal a complicated relationship to urban expansion. On the one hand, economic development requires land investment as space and capital. On the other hand, economic activity agglomeration is an important driving force to a greater urban expansion. The model results show that, while employment has no significant statistical relationship to urban expansion, GDP positively influences the urban expansion process in the Texas Triangle. In contrast, the patent variable as a measurement of technology innovation shows no significant relationship to urban expansion in the Texas Triangle.

As much concern to urban transportation planners, the result shows that the highway density is highly positively related to an urbanized area. However, interestingly, the highway density presents no relationship to a weighted urbanized area. That means the highway density might influence the urban in changing non-urban land to urban land but has little relationship to its intensity. This result provides information for planners to rethinking the use the transportation infrastructure to guide future urban growth. Planners should also consider their role in compact urban development.

## 4. Discussion and Conclusions

This study examines the urban expansion pattern influenced by six major driving factors in the Texas Triangle Megaregion from 2001 to 2016. We first conducted a spatial cluster analysis to explore where the expansion occurred and its magnitude during the period. We then employed a mixed-effect model at the county level to explain the relationships between urban expansion and the focal driving forces, including affiliation to a metropolitan area, population, employment, GDP, technology innovation, and transportation infrastructure. The cluster analysis results show that the urban expansion rate displayed a decreasing trend in the Texas Triangle between 2001 and 2016; 95 percent of new development occurred within the Triangle's metropolitan areas. While clustering patterns varied between the metropolitan areas, the expansion occurred largely in the periphery of central cities. Contrastingly, the urban expansion in non-metropolitan counties was rather scattered. The mixed-effect modeling shows that population, GDP, and highway density were significant predictors of urban expansion.

Between 2001 and 2016, metropolitan areas in the Texas Triangle displayed different patterns despite a shared experience of overall urban expansion. In particular, the Dallas-Houston corridor area showed clustered growth in the periods of 2001–2006 and 2006–2011. This clustered growth pattern, however, did not occur in 2011–2016. The San Antonio-Austin corridor presented clustered expansion throughout the study period of 2001–2016. The two metropolitan areas have now become contiguous, prompting Texas DOT to coordinate joint planning efforts by their respective metropolitan planning organizations [46]. The finding accords with previous studies by demonstrating the heterogeneous expansion patterns at the periphery of large cities [47]. Amid shifting expansion patterns in the areas between large cities and metro areas, state or joint state efforts are necessary to foster cooperation beyond the municipal or agency's jurisdictional boundaries.

Metropolitan planning agencies can play an important role in leveraging regional resource distribution to guide urban expansion [32,48], despite that municipalities make local land use decisions. Improved coordination and cooperation between local and regional entities could lead to desired development outcome. For example, transit-oriented development (TOD) has been widely considered as a tool to facilitate smart urban growth [49]. Since regional transit lines typically traverse multiple municipalities, coordination between MPOs, transit agencies, and local communities is essential to implement TOD strategy at the regional scale. A best-practice example from the Texas Triangle exists from the Dallas region where NCTCOG (North Central Texas Council of Governments), DART (Dallas Area Rapid Transit), City of Dallas as well as other communities along DART routes have coordinated joint efforts to practice TOD in the region [50]. City of Houston initiative Livable Places echoes H-GAC's (Houston-Galveston Area Council) Livable Centers program to promote walkable places and TOD [51,52].

The second regression model estimated in this study considered imperviousness or development intensity, that is, the model of weighted urban land expansion. Adding intensity information into the urban expansion modeling resulted in the loss of statistical significance for the variable highway density. However, the predictor number of patents which was statistically insignificant in the first model, turned to be significant ($p < 0.005$). The contrasting results between the two models suggest that capital investments in highways tend to drive urban expansion horizontally, whereas innovations and technological advances likely push urban expansion vertically towards increased land use efficiency. As concerns over climate change and sustainability grow, local and state governments need to rethink about the conventional strategy of investing in highways to accommodate population and economic growth.

This study has several limitations, suggesting directions for future research. First, this study utilized data on land cover from satellite images, which provide very limited information on land uses for various urban functions. In future research, detailed land use information may be incorporated to allow analyses on variations of urban expansion by different functional types of land uses. Second, this study used imperviousness information from NLCD as a proxy for development intensity; the information provides a rather coarse measure that cannot adequately capture the variation of vertical urban expansion across cities and regions. Lidar data could be used to enhance this study with detailed urban form and vertical development characteristics [53,54]. Lastly, considering the vast area of the Texas Triangle, this study selected county as the geographic unit of analysis. The study findings and discussions are thus limited to the county level. There exist significant within-county variations that this study did not capture. Hence, the study can be refined with use of finer-scale data, for instance, at the census track or block group level.

Despite these limitations, the study results suggest a megaregion-wide emerging trend deviating from the sprawling development course known in Texas' urban growth history. The changing trend can be attributed to the pro-sustainability initiatives taken by several anchor cities and metropolitan planning agencies in the Texas Triangle. Future planning and policy-making efforts should foster this trend toward a sustainable megaregion.

**Author Contributions:** Conceptualization, J.G. and M.Z.; data curation, J.G.; formal analysis, J.G.; funding acquisition, M.Z.; investigation, J.G.; methodology, J.G.; visualization, J.G.; writing—original draft, J.G.; writing—review and editing, M.Z. All authors have read and agreed to the published version of the manuscript.

**Funding:** This research was supported by USDOT UTC Cooperative Mobility for Competitive Megaregions (CM2) (Grant #: 69A3551747135) and the Snell Endowment Grant from the University of Texas at Austin Center for Sustainable Development (CSD).

**Institutional Review Board Statement:** Not applicable.

**Informed Consent Statement:** Not applicable.

**Data Availability Statement:** Data are retrieved from The U.S. Census Bureau (https://www.census.gov/data.html/ accessed on 9 November 2021), Texas Department of Transportation (https://www.txdot.gov/inside-txdot/division/transportation-planning/roadway-inventory.html/ accessed on 9 November 2021), National Landcover Database (https://www.mrlc.gov/data/nlcd-land-cover-conus-all-years/ accessed on 9 November 2021) and Bureau of Economic Analysis (https://www.bea.gov/data/gdp/ accessed on 9 November 2021), and U.S Patent and Trademark Office (https://developer.uspto.gov/data/ accessed on 9 November 2021).

**Acknowledgments:** Thanks to CM2 and CSD for providing funding support to this research. We also want to acknowledge all colleagues, especially Shunhua Bai, in the School of Architecture, University of Texas at Austin for their insightful advice.

**Conflicts of Interest:** The authors declare no conflict of interest. We certify that this submission is original work and is not under review at any other publication. The funders had no role in the design of the study; in the collection, analyses, or interpretation of data; in the writing of the manuscript, or in the decision to publish the results.

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
