# Peer review of "Exploring the Patterns and Drivers of Urban Expansion in the Texas Triangle Megaregion"

_land, doi:10.3390/land10111244_

Round 1

Reviewer 1 Report

Overall, the manuscript aims to answer two questions, including the changes of urban expansion patterns at three different periods in the research region and driver factors of the urban expansion based on the mixed-effect linear regression. The manuscript has solved the above questions. But some issues need to be addressed before the paper will be ready for publication.

  1. This paper attempts to explore which factors play a more important role in the urban expansion process. How do you judge the importance of influencing factors according to the mixed-effect regression? Only by coefficients?

  1. Do independent variables have multicollinearity? I don’t see the analysis about this. It is important for the accuracy of the model result estimation. I suggest the authors add the relevant tests.

  1. Policy and institutional factors are correlates of urban expansion. These factors are also discussed in the article. Why defining institutional factor based on that if a county is within/out of a metropolitan area? I don’t think this indicator is a good proxy for the institutional setting. What indicators are used in the other literature as a proxy for this factor?

  1. Moreover, the industrial structure is also considered as an essential driver factor for urban expansion, but what about that of the Texas Triangle Megaregion?

  1. Is there a lag effect between these independent variables and the urban expansion?

  1. According to the results, “higher growth rates in metropolitan countries than other countries” (line 301), “95% of those expansions occurred in metropolitan areas” (line 303). However, in the table 4, the results of regression tell us “new urban development in the megaregion was found more in non-metro counties than in already well-developed metro counties” (line 366). These conclusions are contradictory.

Reviewer 2 Report

Overall, the findings of this study contribute to improving the understanding of the urban expansion process in a larger study area around the world. However, this manuscript has some other problems that need to be revised:

I think the authors did a fine job in discussing different theories of urban expansion in the Introduction section. However, the authors need to refer to some more case studies related to the subject in the Introduction section. Additionally, in the process of exploring urban expansion patterns and the driving factors, is there any innovation and improvement in the methods used compared to the previous studies? The authors need to discuss this more in detail.

Some basic information about the weather and climate, rainfall, temperature, etc. about the study area should be included in the Study Area section.

What was the basis of selecting the driving factors in this study? Please explain.

Considering the area and the scope of this study, higher spatial resolution data could be used to better quantify the urban expansion process. This should be discussed as a limitation of this study in the manuscript.

I was a little surprised that other important driving factors of urban expansion such as elevation, slope, the existence of physical features like schools, entertainment centers, etc. were not used as driving factors in this study. The process of choosing the drivers should be made clearer.

The term 'Highway Density' should be more clearly specified in terms of which roads were used to calculate the density.

The author used the total population as a driver of urban expansion. However, there might be counties with a smaller area and population size. One approach to minimize this issue is to normalize the population (e.g. population density instead of the total population). Please explain or revise.

The % of urban area change between 2001-2016 is considerably smaller in Beaumont-Port Arthur, Sherman-Denison, and Waco metropolitan compared to other metropolitan areas. Why?

How about the vertical development in the area? The Texas Triangle region grows on the vertical dimension with developments of many new apartments, condominiums, hotels, and restaurants, etc. This can be mentioned as a scope for future research in the manuscript.

The scale unit in Figure 2 is in miles while the unit in other figures is in kilometers. Please use a uniform unit throughout.

The quality of Figures 4 and 6 is poor. Suggest remaking them.

Some other minor comments and suggestions:

Line 34: Urban expansion is often accompanied by...

Line 64-65: This sentence is missing a period (.).

Line 65: should be per capita instead of per capital.

Line 76: why is there an end quotation mark (").

Line 103-104: Please rephrase this sentence.

Line 197: Rephrase this sentence.

Line 225-226: Citations should be properly numbered.

There are some tense problems grammatical errors throughout the manuscript. For example, Line 147 should be "The Texas Triangle had...", Line 202 should be "The highway density was measured...", Line 208 should be "...data were collected... ". Overall grammar should be checked with native speaker.

Reviewer 3 Report

Manuscript ID: land-1430398
Title: Exploring Urban Expansion Patterns and Drivers in the Texas Triangle Megaregion

This manuscript explores the spatiotemporal patterns of urban expansion in the Texas Triangle megaregion and investigates the drivers behind this expansion. The authors used spatial cluster analysis to explore where the expansion occurred, then they used mixed-effect analysis to explain the relationship between the urban expansion and six driving factors. However, the manuscript still needs some improvements to be suitable for publication, especially for maps. My suggestions to develop the manuscript are as follows:

1. In the abstract: the authors mentioned that the urban expansion has decreased over time without mentioning the trend during the period of study.
2. In the introduction section, the authors should discuss the benefits of using spatial cluster analysis and mixed-effect regression comparing to other methods/models used for analyzing driving factors, such as the logistic regression model, Markov chain, and fractal analysis.
3. The results showed a scattered pattern of urban expansion in non-metropolitan areas, comparing to the clustered pattern in metro areas. Please compare these results with the findings of other researchers in similar cities. 
I recommend adding multiple ring buffers to show the change of urban expansion at different distances.
4. The factor of patents is a sign of the prosperity of a region. However, it seems that no relationship between this factor and urban expansion. I guess this factor should be excluded or explain how it can influence urban expansion. Citations of previous work that used this factor are recommended.
5. Lines 322~323, how did you calculate/measure the expansion intensity in the periphery area of the mentioned counties?
After line 328 I recommend discussing the role of the distance factor on urban expansion, especially in the north of the metropolitan.
6. The study recommends using transportation projects as a tool for directing urban expansion in the study area. I suggest adding some examples from similar metropolitan cities
Regarding the figures:
Fig. 1 the study area is so small on the map, please zoom in on the state of taxes to show the urban area of the taxes triangle.
 Fig.2 the map needs some improvements as follows: 
- The roads (major highways) are disconnected, please review.
- The boundary of metropolitan covers some parts of taxes triangle boundary.
- The color of principal cities should be clear (I recommend using red color for them). Other cities can be presented with dark gray color.
- please review the boundary of metropolitan.
Fig. 3 I recommend adding 2 layers to this map for better analysis:
- layer of concentric circles every 2 or 5 km from the centroid of each principal city in the Texas Triangle.
- the layer of roads (highways).
Fig. 4 the legend is unclear.
Fig. 5 please write the title of the X-axis
Fig. 6 the legend is unclear.

General note
Please use the journal format for citations and tables.

Reviewer 4 Report

More information should be added_ "There are two major theories related to the urban expansion mechanism, namely the neoclassical perspective, and the political and institutional perspective. "...it is more complex...

Add Ian McHarg research in this place...since 1960 he has been working to Project with Nature; "However, due to its complicated nature, the field has not reached consensus on which one(s) play a more superior role in the process....

Explain why are similar chinese experiences: "Hebei megaregion in China ..."They thought Shanghai’s urban expansion patterns....

Add references of mountains, and rivers in Fig. 2

Give more details about the method: "We identified hot-spot clusters and spatial outliners of urbanized land at the Census tract level through the Anselin Local Moran’s I statistic....

Explain the criteria for the "Selected Major Drivers of Urban Expansion...why these ones among others

Figure 3, must to be bigger and more with more detail. Add peripherical context, and geographical determinants,... and more explantion of the legend

Same comment to Fig 04. and Fig 06

Define micropolitan, and its differences with metropolitan

Environmental planning and Sustainable Development Goals, looks for the opposite: "That means the highway density might influence the urban in changing non-urban land to urban land but has little relationship to its intensity. This result can provide information for planners to use the transportation infrastructure to guide future urban growth... It has to be re-argumented with present goals

Conclusions should be complemented

Round 2

Reviewer 3 Report

The manuscript has been developed noticeably. However, a couple of comments still exist.
1. The authors discussed the results of the Mixed-effect Regression. However, they did not refer to the fit of the model in the manuscript and explanation of the value of R2 and N.
2. Fig 1 (b) please, change the color of major highways to black to distinguish them from major rivers. In addition, it will be better to change the transparency of the background (topography) to be 70% and add it to the legend of the map. 
Lastly, the location of changes in the response file was different than the actual location in the manuscript. I hardly found the responses. Please write the number of the line after editing the whole manuscript.

Author Response

Thanks for your comments. We revised Figure 1b accordingly. We also explained the r-square and number of observations (N) at the beginning of section 3.2, regression results (Line 1016-1019).

Reviewer 4 Report

The text has been improved.

Author Response

Thank you for providing comments earlier. We further made significant editing regarding our arguments and language.